# Elevated Serum Xanthine Oxidase and Its Correlation with Antioxidant Status in Patients with Parkinson’s Disease

**DOI:** 10.3390/biom14040490

**Published:** 2024-04-18

**Authors:** Ratna Dini Haryuni, Takamasa Nukui, Jin-Lan Piao, Takashi Shirakura, Chieko Matsui, Tomoyuki Sugimoto, Kousuke Baba, Shunya Nakane, Yuji Nakatsuji

**Affiliations:** 1Department of Neurology, Faculty of Medicine, University of Toyama, Toyama 930-8555, Japannukkun@med.u-toyama.ac.jp (T.N.); piaojinlan518@hotmail.com (J.-L.P.); snakane@med.u-toyama.ac.jp (S.N.); 2Research Center for Radioisotope, Radiopharmaceutical, and Biodosimetry Technology, National Research and Innovation Agency, Jakarta 10340, Indonesia; 3Teijin Institute for Bio-Medical Research, Teijin Pharma Ltd., Tokyo 191-8512, Japan; t.shirakura@teijin.co.jp (T.S.);; 4Faculty of Data Science, Graduate School of Data Science, University of Shiga, 1-1-1 Banba, Hikone 522-8533, Japan; tomoyuki-sugimoto@biwako.shiga-u.ac.jp

**Keywords:** xanthine oxidase, Parkinson’s disease, oxidative stress

## Abstract

Parkinson’s disease (PD) is a neurodegenerative movement disorder associated with a loss of dopamine neurons in the substantia nigra. The diagnosis of PD is sensitive since it shows clinical features that are common with other neurodegenerative diseases. In addition, most symptoms arise at the late stage of the disease, where most dopaminergic neurons are already damaged. Several studies reported that oxidative stress is a key modulator in the development of PD. This condition occurs due to excess reactive oxygen species (ROS) production in the cellular system and the incapability of antioxidants to neutralize it. In this study, we focused on the pathology of PD by measuring serum xanthine oxidase (XO) activity, which is an enzyme that generates ROS. Interestingly, the serum XO activity of patients with PD was markedly upregulated compared to patients with other neurological diseases (ONDs) as a control. Moreover, serum XO activity in patients with PD showed a significant correlation with the disease severity based on the Hoehn and Yahr (HY) stages. The investigation of antioxidant status also revealed that serum uric acid levels were significantly lower in the severe group (HY ≥ 3) than in the ONDs group. Together, these results suggest that XO activity may contribute to the development of PD and might potentially be a biomarker for determining disease severity in patients with PD.

## 1. Introduction

Parkinson’s disease (PD) is a neurodegenerative disorder that shows motor symptoms, such as a resting tremor, rigidity, and bradykinesia, and non-motor symptoms, including autonomic dysfunction and sleep disturbances [1]. Although the pathogenesis of hereditary PD is increasingly understood, the pathogenesis of sporadic PD, which is frequently encountered in the clinical setting, remains unclear. A clinical diagnosis can be made without difficulty in typical cases with Parkinsonism, and the progression of symptoms can be monitored by various evaluation scales. However, these are only the surface aspects of the examination and do not reflect the pathophysiology of the disease. Surrogate markers that better reflect the progression of the disease are needed in cases with PD. Various pathological bases have been proposed for PD, as some consider it a syndrome rather than a disease [2]. These include genetics and epigenetics, alpha-synuclein abnormalities, mitochondrial dysfunction, neuroinflammation, altered gut microbiota, neurotransmitter-linked abnormalities, and adenosine receptor abnormalities. In the present study, we focused on mitochondrial dysfunction, which has been proposed for a long time, especially oxidative stress, causing cell damage and death. Oxidative stress has been considered to be strongly linked to the loss of neurons in PD [3]. Accumulating evidence indicates the involvement of oxidative stress in the development and progression of PD [4,5]. Numerous studies have noted an association between the development risk, progression, and severity of PD and low serum uric acid levels [6]. In the metabolism of uric acid, xanthine oxidoreductase (XOR) is known to be widely distributed in major organs, blood vessels, and other tissues in the body as a urate-producing enzyme, and in mammals, XOR is partially modified to act as xanthine oxidase (XO). XO produces uric acid by the reaction “hypoxanthine → xanthine → uric acid” and induces oxidative stress by generating reactive oxygen species (ROS: H_2_O_2_, O^2−^) in the process [7]. Here, we investigated the activity of XO and XOR among oxidative stress, as well as the relationship between these activities and uric acid levels and the severity of PD. Since there have not been many reports on the measurement of XO and XOR activities in the serum from patients with PD [8], we aimed to verify whether XO activity could be the surrogate marker that reflects the pathophysiology and clinical condition of PD.

## 2. Materials and Methods

### 2.1. Ethical Approval

This study was performed in accordance with ethical standards and was approved by the ethics committee of Toyama University Hospital, Japan (Approval No. 29-32). All participants signed informed consent documents.

### 2.2. Reagents

Dimethyl sulfoxide (DMSO), methanol, and potassium carbonate were purchased from Wako Pure Chemical Industries, Ltd., Osaka, Japan. Pterin, Triton X-100, and tributyl phosphate were obtained from Sigma Aldrich, Darmstadt, Germany Tris HCl was purchased from the Nippon gene, Toyama, Japan. Sodium hydroxide solution and centrifugal filter units–PVDF–(0.45 μm) were obtained from Merck, Darmstadt, Germany.

### 2.3. Patients and Samples Collection

Blood samples were collected from 41 patients with PD and 13 patients with other neurological diseases (ONDs) who had visited Toyama University Hospital from 2011 to 2021. PD was diagnosed following the diagnostic criteria of the UK Parkinson’s Disease Society Brain Bank. The cases with ONDs (*n* = 13) consisted of idiopathic normal pressure hydrocephalus (*n* = 7) and cervical spondylosis (*n* = 6). Neuro-degenerative, immunological, and inflammatory diseases were excluded from the current study as pathologies involving oxidative stress; thus, only these diseases remained. We inquired about the history of each of the following: hypertension, diabetes mellitus, chronic kidney disease, thyroid disease, and ischemic heart disease. We evaluated disease severity using the Hoehn and Yahr (HY) scale and calculated a levodopa-equivalent daily dose (LEDD) to account for the dopaminergic agents administered to each patient with PD. When comparing blood uric acid levels between the groups, the patients taking hyperuricemia medications, including allopurinol, and the patients with gout arthritis were excluded. Blood samples were centrifuged at 3000 rpm for 10 min, and the supernatants were transferred to new tubes. Samples were stored at −80 °C until analysis.

### 2.4. Biochemical Analysis

XO activity was determined by measuring the conversion of pterin as a substrate to isoxanthopterin (IXP) in human serum samples. Each sample was reacted with 2% DMSO and 200 μM of pterin at 37 °C for 3 h. After incubation, 4% perchloric acid was added to stop the reactions, followed by centrifugation at 15,000× *g* for 10 min at 25 °C. The resulting supernatant was neutralized with 5 mol/L of K2CO3 and centrifuged at 15,000× *g* for 10 min at 25 °C. Furthermore, the supernatant was added to the centrifugal filters (0.45 μm) and centrifuged at 12,000× *g* for 2 min at 25 °C. The IXP concentration in each sample was measured by high-performance liquid chromatography (HPLC). Serum XO activity was expressed as picomoles of IXP produced per minute per milliliter (pmol/min/mL). The catalase (Invitrogen, Waltham, MA, USA) levels were measured following the manufacturer’s protocol. The uricase and peroxidase method was used for the determination of serum uric acid levels.

### 2.5. Statistical Analysis

Statistical analysis was performed using the JMP^®^ Pro software 2019 ver 15.0 (SAS Institute Inc., Cary, NC, USA). The Wilcoxon test was used for group comparison. Spearman’s test adjusted for age, sex, disease duration, and the age of onset was used to examine the correlation between variables. Fisher’s exact test was used to compare the differences in frequency distribution based on sex, hypertension, diabetes mellitus, chronic kidney disease, current smoking, thyroid disease, and ischemic heart disease. A *p*-value less than 0.05 (*p* < 0.05) was considered statistically significant.

## 3. Results

### 3.1. Clinical Characteristics of Patients with PD and Those with ONDs

The clinical characteristics of the enrolled controls and patients with PD are listed in Table 1. The mean age of patients with ONDs as the control group was 72.7 years. On the other hand, the mean age of patients with PD, age of onset, and disease duration were 70.5 years, 63.1 years, and 7 years, respectively. The median (interquartile range) of HY stages was three (2–3.5). In addition, the mean LEDD received by patients with PD was 459 mg.

### 3.2. Increased XO Activity in Patients with PD

The median (interquartile range) serum XO activity in patients with PD and those with ONDs is shown in Figure 1. We found higher levels of serum XO activity in patients with PD than those with ONDs: 0.23 (0.13–0.52) pmol IXP/min/mL vs. 0.15 (0.06–0.24) pmol IXP/min/mL. Statistical analysis revealed a significant difference (*p* < 0.05) in XO activity between the PD and OND groups. Next, we investigated the correlation between XO activity and PD progression based on HY stages. This progression is related to motor decline and deterioration in the quality of life of patients with PD [9]. We divided the patients with PD into two groups depending on their disease severity: a HY stage less than or equal to two (HY ≤ 2) and a HY stage greater than or equal to three (HY ≥ 3). The demographic of each group of patients with PD is given in Table 2. Statistical analysis revealed a significant difference in serum XO activity between ONDs and PD HY ≥ 3. Furthermore, we used Spearman’s test adjusted for age, sex, disease duration, and the age of onset to examine the correlation between variables. We tested 24 from a total of 41 patients with PD who met these criteria (mild (HY ≤ 2, *n* = 12) and severe (HY ≥ 3, *n* = 12)). The result showed a positive correlation between the serum XO activity and the HY stage (Figure 1C).

### 3.3. Antioxidant Status in Patients with PD

#### 3.3.1. Serum Uric Acid

Uric acid is the end product of purine metabolism derived from the hypoxanthine/xanthine reaction catalyzed by XO [10]. We analyzed serum uric acid levels in patients with PD due to their association with XO activity. In patients with ONDs, one patient taking an anti-hyperuricemic drug was excluded, while in patients with PD, two patients taking anti-hyperuricemic drugs and three patients without any data on serum uric acid were excluded. Figure 2A demonstrates that the median (interquartile range) serum uric acid in patients with ONDs (5.65 (4.4–7.025) mg/dL) was significantly higher than that in patients with PD (4.6 (3.725–5.2) mg/dL. We examined serum uric acid levels according to HY stages. Our result exhibited the median (interquartile range) serum uric acid levels in the ONDs (*n* = 10), PD HY ≤ 2 (*n* = 12), and PD HY ≥ 3 (*n* = 24) groups to be 5.7 (4.4–6.95), 4.95 (3.5–6.7), and 4.4 (3.775–5.175) mg/dL, respectively (Figure 2B). A significant difference was observed in serum uric acid levels between the ONDs group and the PD group in HY ≥ 3 (*p* < 0.05). We then investigated the correlation between serum XO activity and serum uric acid in 36 patients with PD (we excluded three patients with no data and two patients taking an anti-hyperuricemic drug). The results showed that there was no significant correlation between XO activity and serum uric acid (*p* = 0.1855, *p* = 0.2788) (Figure 2C).

#### 3.3.2. Catalase Activity

Catalase is an important antioxidant that breaks down harmful hydrogen peroxide (H_2_O_2_) into oxygen (O_2_) and water (H_2_O) [11]. Since XO generates O^2−^ and H_2_O_2_ during purine catabolism [12], we next analyzed the catalase activity in the serum of patients with PD. There were no significant differences (*p* = 0.146) in catalase activity between the ONDs group with a median (inter-quartile range) of 3.73 (2.79–4.09) and the PD group with a median of 3.25 (2.84–3.70) (Figure 2D). Moreover, there was no significant difference in serum catalase activity according to the HY stage.

## 4. Discussion

Approximately 20% of the oxygen supplied to the body is utilized in the brain, and a significant part of it is transformed into ROS [5,13,14]. This fact is the reason why dopaminergic neurons are unable to avoid oxidative stress. Based on this, we investigated the correlation between serum XO activity and motor dysfunction in PD based on the H and Y stages. Importantly, we found that serum XO activity levels have a positive correlation with disease progression in PD. Several studies demonstrated that O_2_^−^ as a byproduct of XO can be converted into more toxic ROS. O_2_^−^ can react with nitric oxide (NO) to quickly form peroxynitrate (ONOO^−^), which is an extremely reactive oxidizing agent that is more damaging to the cells [9,11,15,16]. Moreover, the interaction between O_2_^−^ and NO can also lead to a reduction in NO bioavailability, which is the main reason for endothelial dysfunction. Excessive ROS production results in oxidative stress, which causes cellular damage to numerous components (lipids, proteins, mitochondria, DNA, and RNA), eventually leading to cell death [10,17]. High levels of oxidative stress and low levels of the antioxidant defense system of the brain are the main factors related to the development of several neurological disorders, such as PD, Alzheimer’s disease, and stroke [11]. Since there are reports of increased oxidative stress in peripheral blood cells in patients with early-stage PD, we assume that PD develops as a result of the progressive degeneration of dopamine neurons in the substantia nigra as a result of increased oxidative stress in the peripheral blood due to aging [18].

In the current study, in contrast to XO activity, uric acid levels as end oxidation products of purine metabolism (adenine and guanine) decreased in the PD group compared with the ONDs group [12]. No significant correlation was found between XO activity and serum uric acid levels. We found significantly lower serum uric acid levels in patients with PD when compared to those with ONDs. A similar correlation between serum uric acid and PD was also reported by Sakuta et al. [19]. Consistent with serum uric acid, significantly lower plasma uric acid was observed in patients with PD than in the controls [20]. In addition, a large population-based cohort study involving 4695 participants found that increased serum uric acid levels were correlated with a reduction in the risk of PD [21]. In addition, it has been reported that, in patients with early PD, the intake of inosine, as a precursor of uric acid, increased blood uric acid levels and inhibited disease progression [22]. It might occur due to oxidative stress being implicated in PD pathogenesis, and uric acid might play a protective role against PD via antioxidant capacity [21]. A low level of serum uric acid in patients with PD, compared to those with ONDs, causes its role as a potent antioxidant to decrease. Therefore, it can be inferred from the present results that it is unable to counter existing free radicals. On the other hand, allopurinol intake reduced blood uric acid levels in PD mouse models but did not affect dopamine neuron degeneration in the substantia nigra [23]. Furthermore, another report showed that there is no association between the use of allopurinol with the progression of PD [24]; the relationship between blood uric acid levels and the pathogenesis of PD is controversial.

Catalase is a crucial antioxidant for cellular defense against oxidative stress because of its ability to convert hydrogen peroxide molecules into harmless molecules (water and oxygen). The deficiency of catalase is closely related to many degenerative diseases, including PD [25]. The mechanism that causes the decrease in catalase activity and a high amount of hydrogen peroxide in PD is likely due to the indirect inhibition of catalase expression by the α-synuclein molecule [25]. Unfortunately, our data showed no significant differences in catalase activity between the ONDs and PD groups.

This study had some limitations. First, the sample size was relatively small. Further study is needed to clarify the relationship between serum XO activity and disease severity in patients with PD by increasing the number of patients with PD, disease controls, and healthy controls. Second, we used the serum from patients with ONDs as a control instead of those from a healthy person. A previous study showed that plasma XO activity in healthy individuals ranged from 0.1 to 0.25 pmol IXP/min/mL [26]. Since the median (interquartile range) serum XO activity in patients with ONDs was 0.15 (0.06–0.24), which is similar to previously reported values in healthy individuals, it is not a problem to use the serum from patients with ONDs as a control.

## 5. Conclusions

Collectively, the results of the present study suggest that XO as a ROS may be a surrogate marker involved in the neurodegenerative process in patients with PD.

## Figures and Tables

**Figure 1 biomolecules-14-00490-f001:**
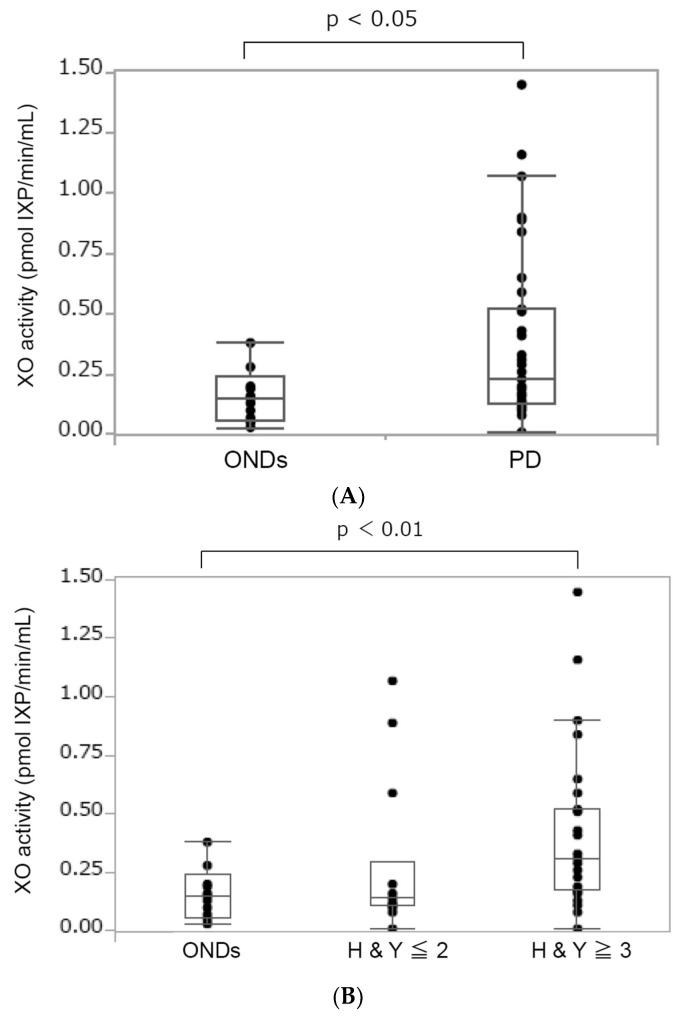
The boxplot of serum XO activity in patients with ONDs and those with PD. (**A**) The boxplot of serum XO activity in patients with other neurological diseases (ONDs) and those with PD. The median (interquartile range) serum XO activity was 0.15 (0.06–0.24) and 0.23 (0.13–0.52) pmol XP/min/mL, respectively. The Wilcoxon test was used to compare the two groups. A *p*-value < 0.05 indicates statistical significance. (**B**) The boxplot of serum XO activity in the patients with ONDs and PD. The median (interquartile range) serum XO activity was 0.15 (0.06–0.24), 0.14 (0.10–0.29), and 0.29 (0.16–0.52) pmol XP/min/mL in patients with ONDs (*n* = 13), PD HY ≤ 2 (*n* = 14), and PD HY ≥ 3 (*n* = 27), respectively. The Wilcoxon test with Holm correction was used to compare the three groups. A value of *p* < 0.05 indicates statistical significance. (**C**) Correlation between CSF ATP levels and the Hoehn and Yahr stages. Spearman’s test adjusted for age, sex, disease duration, and age of onset was used to examine the correlation between variables. r = correlation (95% CI). A *p*-value < 0.05 indicates statistical significance.

**Figure 2 biomolecules-14-00490-f002:**
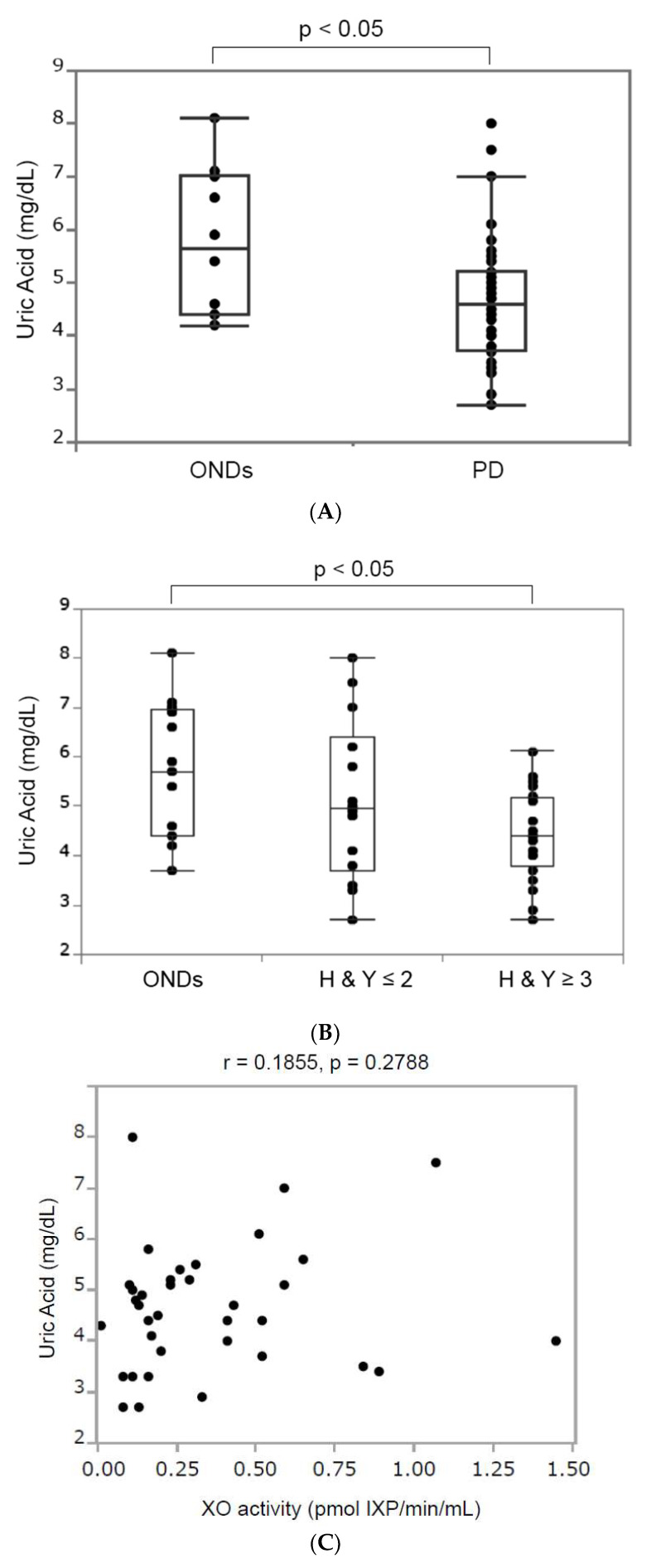
Antioxidant levels in patients with ONDs and those with PD. (**A**) The boxplot of serum uric acid levels in patients with ONDs and those with PD. The median (interquartile range) serum uric acid level in the ONDs group was 5.65 (4.4–7.025), and that in the PD group was 4.6 (3.725–5.2) mg/dL. When comparing serum uric acid, ONDs included 10 patients, and PD included 36 patients. In the ONDs group, three patients taking anti-hyperuricemic drugs were excluded. In patients with PD, two patients taking anti-hyperuricemic drugs and three patients without any data on serum uric acid were excluded (HY ≤ 2 group included 12 patients and HY. The ≥ 3 group included 24 patients). (**B**) The boxplot of serum uric acid levels in patients with ONDs and those with PD. The median (interquartile range) serum uric acid levels were 5.7 (4.4–6.95), 4.95 (3.5–6.7), and 4.4 (3.775–5.175) pmol XP/min/mL in patients with ONDs, PD HY ≤ 2 (*n* = 12), and PD HY ≥ 3 (*n* = 24), respectively. The Wilcoxon test with Holm correction was used for comparing the three groups. A *p*-value < 0.05 indicates statistical significance. (**C**) Correlation between serum XO activity and serum uric acid in 36 patients with PD (r = 0.1855, *p* = 0.2788). (**D**) The boxplot of serum catalase in patients with ONDs and those with PD. There were no significant differences (*p* = 0.146) in catalase activity between the ONDs group with a median (inter-quartile range) of 3.73 (2.79–4.09) and the PD group with a median of 3.25 (2.84–3.70). The Wilcoxon test was used to compare the two groups. A *p*-value < 0.05 was considered statistically significant.

**Table 1 biomolecules-14-00490-t001:** Clinical characteristics of patients with PD and ONDs.

	ONDs (*n* = 13)	PD (*n* = 41)	*p* Value
Male	7 (53.9)	22 (53.7)	1
Age (years)	72.7 ± 9.0	70.5 ± 9.9	0.62
Age of onset (years)		63.1 ± 10.4	
Disease duration (years)		7.0 ± 5.5	
Hoehn and Yahr stages		3 (2–3.5)	
LEDD (mg)		459 ± 305	

Data are presented as the mean ± SD and median (interquartile range) or number (%), *n* = number of samples. ONDs: other neurological diseases (cervical spondylosis, idiopathic normal pressure hydrocephalus). PD: Parkinson’s disease. LEDD; levodopa equivalent daily dose.

**Table 2 biomolecules-14-00490-t002:** Demographics of PD patients based on Hoehn and Yahr (HY) stages.

	ONDs (*n* = 13)	HY ≤ 2 (*n* = 14)	HY ≥ 3 (*n* = 27)	*p* Value
Male	7 (53.9)	8 (57.1)	14 (51.9)	0.95
Age (years)	72.7 ± 9.0	67.0 ± 12.2	72.3 ± 8.1	0.52
Age of onset (years)		62.5 ± 13.1	63.6 ± 9.4	0.93
Disease duration (years)		3.9 ± 3.0	8.7 ± 5.8	<0.05 *
LEDD (mg)		332 ± 266	524 ± 308	0.05
HT	10 (76.9)	5 (35.7)	8 (29.6)	<0.05 *
DM	2 (15.4)	2 (14.3)	4 (14.8)	1
CKD	3 (23.1)	4 (28.6)	6 (22.2)	0.9
Current smoking	2 (15.4)	0 (0)	0 (0)	<0.05 *
Thyroid disease	3 (23.1)	0 (0)	2 (7.4)	0.11
Ischemic heart disease	0 (0)	0 (0)	1 (3.7)	0.6

Data are presented as the mean ± SD and number (%). HT: hypertension, DM: diabetes mellitus, and CKD: chronic kidney disease (eGFR < 60 mL/min/1.73 m^2^). * Indicates the Wilcoxon test or Fisher’s exact test.

## Data Availability

Data are contained within the article.

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
