# Peer review of "Elevated Serum Xanthine Oxidase and Its Correlation with Antioxidant Status in Patients with Parkinson’s Disease"

_biomolecules, 2024, doi:10.3390/biom14040490_

Round 1

Reviewer 1 Report

Comments and Suggestions for Authors

I am not convinced by the novelty of the findings reported in this manuscript: in fact, both an increase of XO (quoted ref. 8) (with no alteration of the levels of catalase) and decreased blood levels of uric acid have been reported in PD patients. Thus, this seems to be only a confirmatory study.

1-The sample size is abnormally low (41+13 patients). I would need to see an explicit power analysis to accept such a low number of patients.

2-I would expect that OND group should include other diseases that are often differentially diagnosed in PD (e.g. essential tremor, dementia with Lewis bodies, multiple system atrophy, dystonia or CBS). In other words, the OND group is a priori biased and does not make full clinical sense (at least to me).

3-Some infos in the methods are required, namely: i) Was the consumption of allopurinol controlled in the patients? ii) Did the patients suffer from arthritis?

4-I cannot understand the relation between oxidative stress in dopaminergic neurons and the levels of Xo in the blood. It is for me very difficult to understand how a few dopaminergic neurons can account for increased the levels of XO in the blood. In other words, I foresee no evidence supporting that dopamine neurons are particular contributors of the levels of XO in the blood.

5-Also, I am not aware of any evidence that there is any objective experimental demonstration of a relation between redox stress in the nigra and blood markers such as XO or urid acid.

6-It is surprising that the SURE-PD trials related to rid acid were even mentioned (JAMA Neurol. 2014 Feb;71(2):141-50. doi: 10.1001/jamaneurol.2013.5528) and they certainly should.

7-I was also expecting to see a comment in the Discussion on the relation (or lack of thereof) between the intake of allopurinol and PD.

Author Response

  • The sample size is abnormally low (41+13 patients). I would need to see an explicit power analysis to accept such a low number of patients.

→ The Reviewer 1 is correct in pointing out that this is a concern of ours. This is the first paper to show a positive correlation between the severity of Parkinson's disease and blood oxidative stress, but it is only a preliminary study and needs to be validated in a larger number of cases in the future.

  • I would expect that OND group should include other diseases that are often differentially diagnosed in PD (e.g. essential tremor, dementia with Lewis bodies, multiple system atrophy, dystonia or CBS). In other words, the OND group is a priori biased and does not make full clinical sense (at least to me).

→ In this study, blood XO activity was not measured in patients with other neurodegenerative diseases like multiple system atrophy (MSA) and corticobasal syndrome (CBS). Also, to the best of our knowledge, we could not find previous reports that measured blood XO in these patients. So, as the Reviewer 1 pointed out, blood XO activity may not be useful as a diagnostic marker. However, we showed that blood XO activity correlates with the severity of PD, and we think that blood XO activity may be a useful severity marker in patients with PD. As noted in the discussion, the blood XO activity of OND group is similar to that of healthy persons in previous reports, and we think that it is not a problem to use the serum from patients with ONDs as a control.

  • Some infos in the methods are required, namely: i) Was the consumption of allopurinol controlled in the patients? ii) Did the patients suffer from arthritis?

→ In this study, the patients taking allopurinol were excluded when comparing uric acid levels between groups. Also, there were no patients with gout arthritis. We have corrected the method(line 4 on page 6).

  • I cannot understand the relation between oxidative stress in dopaminergic neurons and the levels of Xo in the blood. It is for me very difficult to understand how a few dopaminergic neurons can account for increased the levels of XO in the blood. In other words, I foresee no evidence supporting that dopamine neurons are particular contributors of the levels of XO in the blood.
  • Also, I am not aware of any evidence that there is any objective experimental demonstration of a relation between redox stress in the nigra and blood markers such as XO or urid acid.

→ As the Reviewer 1 pointed out, it is unlikely that increased oxidative stress in the substantia nigra affects the levels of XO in the pheriperal blood. Since there are report of increased oxidative stress in peripheral blood cells in patients with early-stage PD, we assume that PD develops as a result of progressive degeneration of dopamine neurons in the substantia nigra as a result of increased oxidative stress in the peripheral blood due to aging. We have modified the Discussion (line 15 on page 11).

  • It is surprising that the SURE-PD trials related to rid acid were even mentioned (JAMA Neurol. 2014 Feb;71(2):141-50. doi: 10.1001/jamaneurol.2013.5528) and they certainly should.

→ Thank you for pointing this out. I have added a discussion and cited the reference to the fact that inosine intake increases blood uric acid level and inhibits the progression of PD (line 9 on page 12).

  • I was also expecting to see a comment in the Discussion on the relation (or lack of thereof) between the intake of allopurinol and PD.

→ Currently, we think that the relationship between taking allopurinol and the development of PD is controversial. We have modified the discussion (line 14 on page 12).

Reviewer 2 Report

Comments and Suggestions for Authors

Ratna Dini Haryuni and co-workers studied the serum xanthine oxidase and catalase activities as well as the uric acid levels in Parkinson’s disease patients with various disease severity. The manuscript is of some interest, however, there are some questions/comments.  

First of all, the authors used samples from patients with other neurological diseases (ONDs) as controls, but excluded neurodegenerative, immunological, and inflammatory diseases as the pathologies of those disorders involve oxidative stress (lines 76-81). However, it raises the question whether the observed alterations are characteristic indeed for Parkinson’s disease, or, more, generally, for diseases with oxidative stress. In that case, XO activity might not be a suitable surrogate marker for Parkinson’s disease.

Fig. 1B: In the figure, there are different H and Y stages, although according to lines 127-129: “We divided the patients with PD into two groups depending on their disease severity: HY stage less than or equals 2 and HY stage greater than or equals 3.”  The figure legend is also incorrect.

Discussion lines 217-218: “this study discovered a significant decrease in serum uric acid levels in males of the PD group when compared with the control group”. Where is a figure/table supporting this observation?

Author Response

  • First of all, the authors used samples from patients with other neurological diseases (ONDs) as controls, but excluded neurodegenerative, immunological, and inflammatory diseases as the pathologies of those disorders involve oxidative stress (lines 76-81). However, it raises the question whether the observed alterations are characteristic indeed for Parkinson’s disease, or, more, generally, for diseases with oxidative stress. In that case, XO activity might not be a suitable surrogate marker for Parkinson’s disease.

→ In this study, blood XO activity was not measured in patients with other neurodegenerative diseases like multiple system atrophy (MSA) and progressive supranuclear palsy (PSP). Also, to the best of our knowledge, we could not find previous reports that measured blood XO in these patients. So, as the Reviewer 2 pointed out, blood XO activity may not be useful as a diagnostic marker. However, we showed that blood XO activity correlates with the severity of PD, and we think that blood XO activity may be a useful severity marker in patients with PD.

  • 1B: In the figure, there are different H and Y stages, although according to lines 127-129: “We divided the patients with PD into two groups depending on their disease severity: HY stage less than or equals 2 and HY stage greater than or equals 3.”  The figure legend is also incorrect.

→ Thank you for pointing this out. We examined the relationship between the severity of PD and serum XO, both by dividing patients into two groups and by correlating with H&Y. The data from the study with two groups of patients with PD were added as Fig 1B, and the text and Fig legend has been modified (line 18 on page 8, line 5 on page 9, Fig legends).

  • Discussion lines 217-218: “this study discovered a significant decrease in serum uric acid levels in males of the PD group when compared with the control group”. Where is a figure/table supporting this observation?

→ As the Reviewer 2 pointed out, the present study did not compare blood uric acid levels by sex in patients with PD and ONDs. We have removed the sentence (line 6 on page 12).

Round 2

Reviewer 1 Report

Comments and Suggestions for Authors

After reading the revised manuscript, especially the first paragraph of the Discussion, I am still left with the idea that an eventual oxidative stress in the nigra is mixed with the measured alteration of oxidative stress in the plasma, whereas there is no evidence or rationale to link the two compartments.

The increased levels of uric acid in the control group may be related with the increase of adenosine plasma levels associated with hydrocephalus, a condition afflicting most of the patients in the OND control group.

-l.35-36: a verbe is missing in the sentence.

Author Response

After reading the revised manuscript, especially the first paragraph of the Discussion, I am still left with the idea that an eventual oxidative stress in the nigra is mixed with the measured alteration of oxidative stress in the plasma, whereas there is no evidence or rationale to link the two compartments.

The increased levels of uric acid in the control group may be related with the increase of adenosine plasma levels associated with hydrocephalus, a condition afflicting most of the patients in the OND control group.

→ As the Reviewer 1 pointed out, this study does not clarify how increased blood oxidative stress is related to increased central nervous system oxidative stress. Also, in comparing blood uric acid levels, patients with idiopathic normal pressure hydrocephalus (iNPH) may not be appropriate controls. We also believe that these important issues should be resolved in further research.

-l.35-36: a verbe is missing in the sentence.

→ Thanks for pointing that out. We have corrected the introduction (line 2 on page 3)

Reviewer 2 Report

Comments and Suggestions for Authors

The authors have answered my questions/comments.

Author Response

The authors have answered my questions/comments.

→ Thank you very much for providing important comments. We are thankful for the time and energy you expended.